# Trends in US pediatric mental health clinical trials: An analysis of ClinicalTrials.gov from 2007–2018

Joshua R. Wortzel[1☯]*, Brandon E. Turner[2☯], Brannon T. Weeks[3], Christopher Fragassi[1‡], Virginia Ramos[1‡], Thanh Truong[4‡], Desiree Li[4‡], Omar Sahak[4‡], Thomas G. O'Connor[1]

1 Department of Psychiatry, University of Rochester, Rochester, NY, United States of America,
2 Department of Radiation Oncology, MGH, Harvard University, Boston, MA, United States of America,
3 Department of Gynecology and Obstetrics, MGH, Harvard University, Boston, MA, United States of America, 4 Department of Psychiatry and Behavioral Sciences, Stanford University, Stanford, CA, United States of America

☯ These authors contributed equally to this work.
‡ CF, VR, TT, DL and OS also contributed equally to this work.
* jrwortzel@gmail.com

**Data Availability Statement:** All data and code files are available from Mendeley Data repository at http://dx.doi.org/10.17632/pgckc382cr.1.

## Abstract

Whereas time trends in the epidemiologic burden of US pediatric mental health disorders are well described, little is known about trends in how these disorders are studied through clinical research. We identified how funding source, disorders studied, treatments studied, and trial design changed over the past decade in US pediatric mental health clinical trials. We identified all US pediatric interventional mental health trials submitted to ClinicalTrials.gov between October 1, 2007 and April 30, 2018 (n = 1,019) and manually characterized disorders and treatments studied. We assessed trial growth and design characteristics by funding source, treatments, and disorders. US pediatric mental health trials grew over the past decade (compound annual growth rate [CAGR] 4.1%). The number of studies funded by industry and US government remained unchanged, whereas studies funded by other sources (e.g., academic medical centers) grew (CAGR 11.3%). Neurodevelopmental disorders comprised the largest proportion of disorders studied, and Non-*DSM-5* (*Diagnostic and Statistical Manual-5*) conditions was the only disorder category to grow (14.5% to 24.6%; first half to second half of decade). There was significant growth of trials studying non-psycho/pharmacotherapy treatments (33.8% to 49.0%) and a decline in trials studying pharmacotherapies (31.7% to 20.6%), though these trends differed by funding source. There were also notable differences in funding sources and treatments studied within each disorder category. Trials using double blinding declined (26.2% to 18.0%). Limitations include that ClinicalTrials.gov is not an exhaustive list of US clinical trials, and trends identified may in part reflect changes in trial registration rather than changes in clinical research. Nevertheless, ClinicalTrials.gov is among the largest databases available for evaluating trends and patterns in pediatric mental health research that might otherwise remain unassessable. Understanding these trends can guide researchers and funding bodies when considering the trajectory of the field.

**Funding:** The authors received no specific funding for this work.

**Competing interests:** The authors have declared that no competing interests exist.

## Introduction

Time trend data are fundamental to epidemiological research [1], and they are widely studied in psychiatry and psychology [2, 3]. Increases in the prevalence of pediatric mental health conditions are significant and extend to multiple psychiatric disorders [4]. For example, in the 1960s, one in 2,500 children was diagnosed with autism [5], yet by 2014, this number was as high as one in every 59 children in the United States [6]. Current estimates of the prevalence of attention deficit hyperactivity disorder (ADHD) are between 4–12% in school aged children, representing a 24% increase since 2001 [7]. The number of US children diagnosed with either depression or anxiety has also increased from 5.4% in 2003 to 8.4% in 2012 [8]. The extent to which these trends reflect changes in assessment tools and diagnostic sensitivity is subject to some debate; however, in contrast to the epidemiology, little is known about accompanying time trends in pediatric mental health clinical research.

A reliable source of information on time trends in clinical research is the National Institutes of Health's ClinicalTrials.gov registry, which was created in 2000 [9]. ClinicalTrials.gov has become one of the largest registries for clinical research internationally, and it currently contains detailed information on more than 365,000 clinical studies conducted in over 210 countries. Over 300 research articles have utilized ClinicalTrials.gov to characterize trends in clinical research, including studies assessing trends in trial design, trial funding, and disorders and treatments studied [10–17].

There have been several studies that have utilized the ClinicalTrials.gov registry to identify trends in mental health trials [14–17]. In their analysis of all mental health trials in the registry from 2007 to 2014, Arnow and colleagues reported that universities and hospitals funded the majority of mental health trials, that most industry-funded trials studied pharmacotherapies, and that government-funded studies targeted behavioral interventions more than pharmacotherapies [16]. Similarly, in their analysis of mental health clinical research from 2007 to 2018, Wortzel and colleagues found a significant decline in funding from industry and US government sources and a significant increase in funding from academic medical centers and hospitals. A decline in the proportion of mental health trials using blinding and oversight by data monitoring committees was also noted, which occurred in the context of an increasing proportion of trials studying behavioral and non-pharmacological interventions. In addition, there was significant growth of trials studying Non-*DSM-5* (*Diagnostic and Statistical Manual-5*) conditions.

While Wortzel and colleagues identified that over 16% of US mental health trials registered in ClinicalTrials.gov were conducted in pediatric populations, this study did not further parse trends in this patient population [17]. There have been a limited number of studies that have identified trends in pediatric mental health clinical trials, and these have focused on trends within the published literature concerning specific treatment types. For example, systematic reviews have shown that an increasing number of trials have studied mobile apps and resilience-focused school-based interventions to treat psychological distress and wellbeing in children [18, 19]. Meta-analyses have also explored the growth of trials studying mindfulness techniques in the treatment of child and adolescent mental health and cognitive disorders [20, 21]. These reviews provide valuable perspectives on clinical research developments within each of these specific subfields of pediatric mental health; however, they do not identify larger trends across all pediatric mental health clinical research, such as changes in funding, disorders studied, or how the research in each subfield has changed relative to others being studied. Considering that nearly 50% of pediatric trials are discontinued or remain unpublished within 58 months of trial completion, these analyses of published data are also likely missing important nuances in the trends of pediatric mental health clinical research being conducted [22].

Therefore, fundament gaps still exist in our understanding of current trends in pediatric mental health trials, and these might be answered through an analysis of a large, national, clinical trials registry, such as ClinicalTrials.gov.

In the current study, we used a similar, established methodology implemented by Wortzel and colleagues to examine time trends in clinical trials specific to pediatric mental health [17]. We evaluated trends in the funding, disorders and treatments studied, and trial design characteristics of US pediatric mental health trials registered in ClinicalTrials.gov from 2007 to 2018 and discuss their significance.

## Materials and methods

### Data selection and classification

Records were downloaded on April 30, 2018 for all 274,029 trials submitted to ClinicalTrials. gov as of April 30, 2018 using the Aggregate Analysis of ClinicalTrials.gov (AACT), a relational database of publicly available ClinicalTrials.gov data [23]. Trials submitted to the registry on or after October 1, 2007 were selected to coincide with the passing of the Food and Drugs Administration Amendments Act (FDAAA) on September 27, 2007, which stipulated that all United States non-phase 1 trials involving US Food and Drug Administration (FDA) regulated drug and biological products, as well as non-feasibility trials of FDA regulated devices, were mandated to report to a clinical trials registry [10]. We further selected trials labeled interventional in the registry to correspond to clinical trials in which participants were assigned to receive interventions, pharmacological or non-pharmacological, based on a protocol [14]. A psychiatrist reviewed the list of all Medical Subject Headings (MeSH) and Disease Condition terms in the ClinicalTrials.gov registry, and those terms deemed relevant to mental health were selected and reviewed by another physician. The full list of the MeSH and Disease Condition terms used for this analysis has been published elsewhere [17]. The trials utilizing these terms were divided among six psychiatrists who manually reviewed the official title and study descriptions to (i) identify trials relevant to pediatric mental health (i.e., the trial description discussed studying 'children', 'adolescents', or patients ≤18 years old), (ii) categorize these trials according to the disorder index categories in the Section II Diagnostic Criteria and Codes of the *DSM-5* [17, 24], and (iii) categorize by treatment type. A sample of 250 trials was reviewed by all six psychiatrists to ensure agreement on the labeling criteria. Trial categorizations with any ambiguity were marked and then reviewed and clarified by another psychiatrist. Because requirements for trial registration vary significantly by country, only trials with research sites exclusively within the United States were included in this analysis.

### Changes to the initial protocol

We developed the original protocol for our analysis in April 2019. This protocol was not pre-registered. The original analysis was conducted in May 2019. However, after receiving reviewers' feedback for a separate analysis of the portfolio of all mental health clinical trials registered in ClinicalTrials.gov [17], we subsequently applied these suggestions to our analysis in pediatric mental health trials and modified our protocol accordingly. These changes to the protocol are detailed in S1 Table (items 1–4). In brief, we made four changes. First, while we initially analyzed both US and international studies, we ultimately decided to limit the analysis to only US trials, as has been previously published [10, 13, 25]. This is because trial registration practices differ significantly by country, and there was concern that inclusion of international trials would confound our results (i.e., observed trends could be due to true regional differences in trial characteristics or differences in regional trial registration). Second, our initial analysis excluded the ClinicalTrials.gov funder category US Fed, as has been previously published [14],

because the US Fed category comprises only 3.5% of trials in the registry. However, we subsequently combined US Fed-funded trials with NIH-funded trials to form a new funder category called 'US Govt' to better capture changes in US government-funded trials. Third, we initially clustered Phase 1/2 and Phase 2/3 trials under the phase category 'Not Applicable;' however, in our revised analysis we grouped these trials with Phase 2 and Phase 3 trials, respectively, as these trials were deemed to have ultimately reached Phase 2 and Phase 3 status. Fourth, we included the citations of Arnow and colleagues and Wortzel and colleagues in our revised protocol, as these analyses contextualized our study and motivated several changes made to the revised analysis [16, 17]. The revised protocol was created in April 2020, and the revised analysis in accordance with this protocol was completed in May 2020.

After incorporating feedback from reviewers of this manuscript, we made two additional changes to the protocol in January 2021 [16, 17]. These are also detailed in S1 Table (items 5–6). First, there was concern that two of the terms used to label trial treatments were unclear and contributed to confusion when interpreting the results. The term 'Interventional' was changed to 'Stimulation' to describe electroconvulsive therapy, deep brain stimulation, and transcranial magnetic stimulation. The term 'Alternative' was changed to 'Non-Psycho/Pharmacotherapy' to describe interventions that did not fall into the categories of 'Pharmacotherapy', 'Psychotherapy', or 'Stimulation'. Second, in the original analysis, an alpha threshold of 0.01 was used. However, a more stringent threshold of $\alpha = 0.005$ was used in the revision, as has been previously published [26].

### Trial characteristics

We analyzed each trial on 11 dimensions:

1. Date of submission (dates ranged from October 1, 2007 to April 30, 2018). We divided our 127-month study period at the approximate midpoint into a 63-month 'Early' period (October 1, 2007 to December 31, 2012) and a 64-month 'Late' period (January 1, 2013 to April 30, 2018). Time of submission was assessed as a dichotomous variable using these groupings to look at proportional changes in trial characteristics. Monotonic growth trends were assessed by grouping trials by year of submission. All year-to-year analyses included only years with a full 12-month collection of data (i.e., 2008–2017).

2. Trial primary objective (categories included 'Treatment', 'Prevention', 'Supportive Care', and 'Other'). 'Treatment', 'Prevention', and 'Supportive Care' were categories taken directly from the categorization in ClinicalTrials.gov. 'Other' was generated by combining the category Other in ClinicalTrials.gov with the categories Diagnostic, Health Services Research, Screening, and Basic Science, which together comprised 10.0% of trials. 'Treatment' denotes trials in which one or more interventions were assessed to treat a disease, syndrome, or condition. 'Prevention' denotes trials in which one or more interventions were evaluated to prevent the development of a specific disease or health condition. 'Supportive Care' denotes trials in which one or more interventions were examined to maximize comfort, minimize side effects, or mitigate decline in participants' health or function.

3. Trial phase (categories included 'Phase 1', 'Phase 1/2–2', 'Phase 2/3–3', 'Phase 4', and 'Not Applicable'). 'Phase 1' was generated by grouping the ClinicalTrials.gov categories Early Phase 1 and Phase 1. 'Phase 1/2–2' was generated by grouping the ClinicalTrials.gov categories Phase 1/2 and Phase 2. 'Phase 2/3–3' was generated by grouping the ClinicalTrials.gov categories Phase 2/3 and Phase 3. 'Phase 4' and 'Not Applicable' were taken directly from these corresponding categories in ClinicalTrials.gov. Of note, 'Not Applicable' does not

refer to missing data but rather to trials without FDA-defined phases, such as trials studying devices or behavioral interventions.

4. Number of arms (grouped by range: 'One', 'Two', or '≥Three'). Number of arms, as reported in ClinicalTrials.gov, were grouped and treated as nominal variables using these categories.

5. Blinding (categories included 'None', 'Single', and 'Double'). The category 'Blinding' was generated from the category Masking in ClinicalTrials.gov.

6. Use of randomization (category included 'Yes'). This was taken directly from the categorization in ClinicalTrials.gov.

7. Oversight by a data monitoring committee (DMC) (category included 'Yes'). This was taken directly from the categorization in ClinicalTrials.gov.

8. Number of sites (categories included 'One', 'Two', 'Three–Ten', and '>Ten'). Number of sites, as reported in ClinicalTrials.gov, were grouped and treated as nominal variables using these categories.

9. Funding source (categories included 'Industry', 'United States Government [US Govt]', and 'Academic Medical Centers/Hospitals/Others [AMC/Hosp/Oth]'). The category 'Industry' was taken directly from the categorization in ClinicalTrials.gov. The category 'US Govt' was generated from the ClinicalTrials.gov categories NIH and US Fed, as previously described [16, 17]. The 'Other' category is primarily composed of academic institutions or hospitals, and the minority were charities and foundations, which is why the label 'Academic Medical Centers/Hospitals/Others' is used [16, 17]. We used a hierarchical funder designation, such that trials with any industry involvement were labeled 'Industry', any remaining trials with US government involvement were labeled 'US Govt', and all remaining trials were labeled 'AMC/Hosp/Oth'. This method was used to capture the influence of industry and government on trial characteristics, as has been previously published in analyses of the registry [12–14, 17].

10. Treatment type (categories include 'Non-Psycho/Pharmacotherapy', 'Stimulation', 'Pharmacotherapy', and 'Psychotherapy'). These categories were created manually to identify the treatments studied, and they were further divided into subcategories. For 'Pharmacotherapy', agents tested were grouped according to drug class (e.g., stimulants, antidepressants, antipsychotics, etc.). For 'Psychotherapy', therapies were grouped by type (e.g., dialectical behavioral therapy, cognitive behavior therapy, etc.). Trials were labeled 'Stimulation' if they studied deep brain stimulation, transcranial magnetic stimulation, or electroconvulsive therapy. Trials with treatments that did not fit into these three categories were labeled 'Non-Psycho/Pharmacotherapy', which was broken down into subcategories including 'Technology' (e.g., interactive phone applications, videogames, etc.), 'Telecommunication' (e.g., telepsychiatry, teletherapy, etc.), 'Community Programs' (e.g., after-school programs, school-wide substance use campaigns, etc.), 'Community Outreach' (e.g., assertive community treatment teams, integrating mental health services into pediatric outpatient primary care centers, etc.) 'Diet and Exercise' (e.g., nutritional supplements, exercise programs, etc.), 'Mediation and Yoga', and 'Basic Science' (e.g., fMRI, genetic profiling, biomarkers, etc.). Trials were assigned more than one treatment category (e.g., when pharmacotherapy was compared to or used as an adjunct to psychotherapy) or subcategory (e.g., a phone app with guided meditations) when appropriate, and consequently the percentages of trials by treatment category and subcategory sum to greater than 100%.

11. Disorder categories (categories include 'Anxiety', 'Bipolar', 'Depression', 'Disruptive, Impulse Control, & Conduct', 'Dissociative', 'Feeding & Eating' 'Gender Dysphoria', 'Neurocognitive', 'Neurodevelopment', 'Obsessive-Compulsive', 'Paraphilic', 'Personality', 'Schizophrenia Spectrum', 'Sexual Dysfunction', 'Sleep-Wake', 'Somatic Symptom', 'Substance & Addiction', 'Trauma & Stressor', and 'Non-*DSM-5* Conditions'). These categories were created manually to identify the disorders studied. Trials that identified disorders by *DSM-IV or–IV-R* diagnostic nomenclature were reclassified using equivalent terms in the *DSM-5*. Trials that did not clearly match any *DSM-5* categories were marked 'Non-*DSM-5* Conditions.' Given the significant number of trials (n = 206) in this category, we further subcategorized the 'Non-*DSM-5* Conditions' (S2 Table). Trials were labeled with as many categories as were relevant, and consequently the percent of trials by disorder category sums to greater than 100%.

## Statistical analysis

We analyzed trial data using descriptive statistics. Because certain fields are optional in ClinicalTrials.gov, approximately 5% of trials had missing data, and, consequently, the total number of trials varies slightly between fields. When data were missing, these trials were excluded. The sample size for each trial characteristic is reported in the tables to note when trial number varies due to exclusion of trials with missing data. We assessed for differences between the distributions of categorical variables of trial characteristics using two-sided Pearson $\chi^2$ tests. We assessed the statistical significance of monotonic trends over time (i.e., compound annual growth rates [CAGR]) using post-hoc Mann-Kendall tests to test the null hypothesis that the number of trials did not change over time. While there are no specific reporting guidelines that have been developed for this type of analysis of trial registries, we adhered to the Strengthening the Reporting of Observational Studies in Epidemiology (STROBE) reporting guidelines for cross-sectional studies [27].

Due to the number of effects explored, we focused on results which achieved statistical significance at the α = 0.005 level, as has been previously published [26]. All analyses were performed using the R statistical programming language, version 3.5.0 [28]. We used the following R packages: Tidyverse [29], Ggpubr [30], Kendall [31], and Coin [32].

## Results

### Study selection

There were 274,416 clinical trials registered in ClinicalTrials.gov as of April 30, 2018. We excluded 56,145 trials because they were not interventional (i.e., participants did not receive interventions based on a protocol), and we excluded 38,109 trials because they were submitted prior to October 1, 2007 (i.e., prior to the passing of the FDAAA). Of the remaining 180,162 interventional trials within this time period, we identified 11,176 trials relevant to mental health, and 6,302 of these trials were conducted within the United States. Of these trials, we identified 1,019 US interventional pediatric mental health trials, which comprised 62.7% (1,019/1,626) of global pediatric mental health interventional trials and 16.2% (1,019/6,302) of all US mental health interventional trials in the registry from October 1, 2007 to April 30, 2018 (Fig 1).

### Growth of trials and trial characteristics over time

From 2008 to 2017, the annual number of US pediatric mental health trials increased (CAGR 4.1%, p = 0.0030) (Fig 2A). Annual growth of US pediatric mental health trials differed by

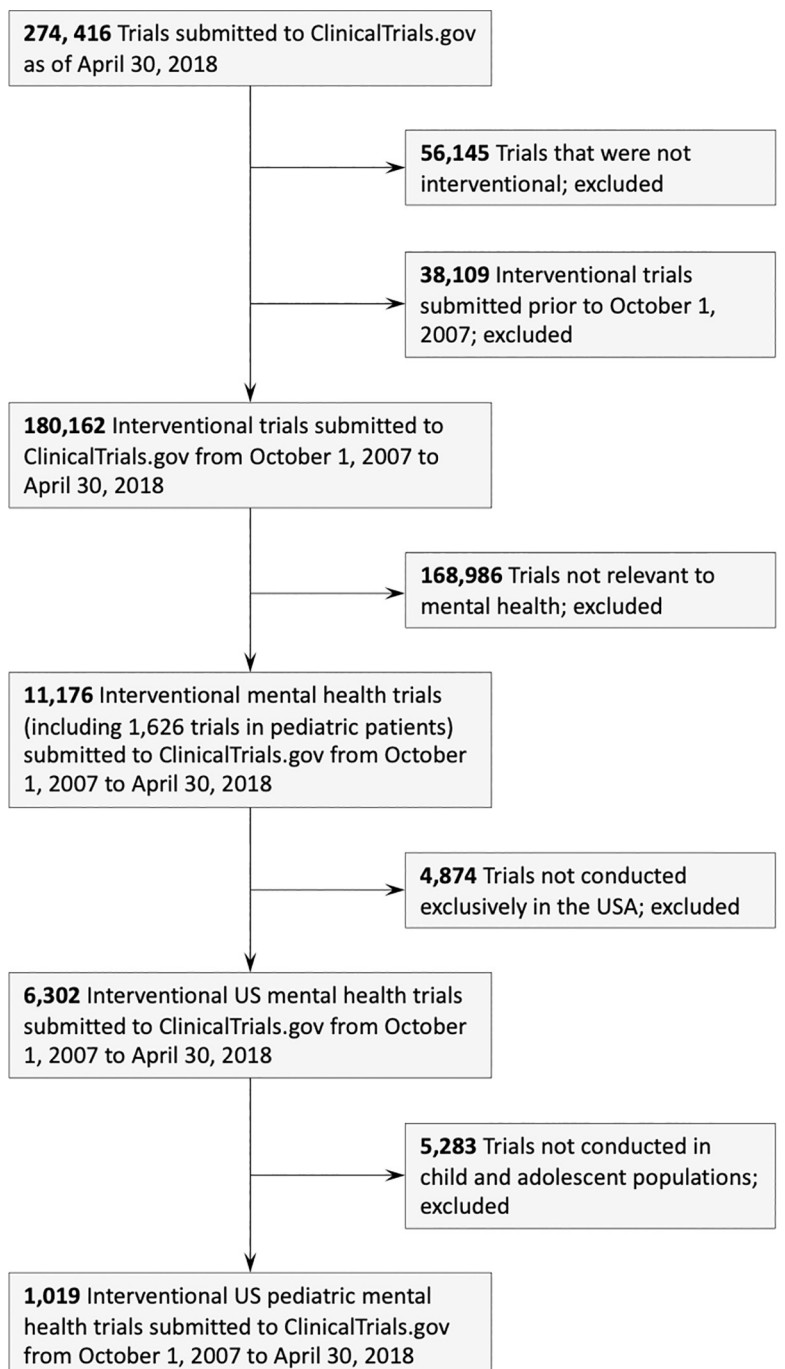

**Fig 1. A flow diagram of inclusion of US pediatric interventional mental health trials registered in ClinicalTrials. gov.**

funding source (Fig 2B), and the proportion of trials also differed by funding source when the data were stratified into early (2007–2012) and late (2013–2018) time periods (Table 1). Annual growth of US government-funded trials trended downward (CAGR -2.6%, p = 0.10) and proportionally decreased between the early and later periods (172 to 157 trials, 39.0% to 27.2%, p<0.0001). Annual growth of industry-funded trials was not monotonic (CAGR -3.3%,

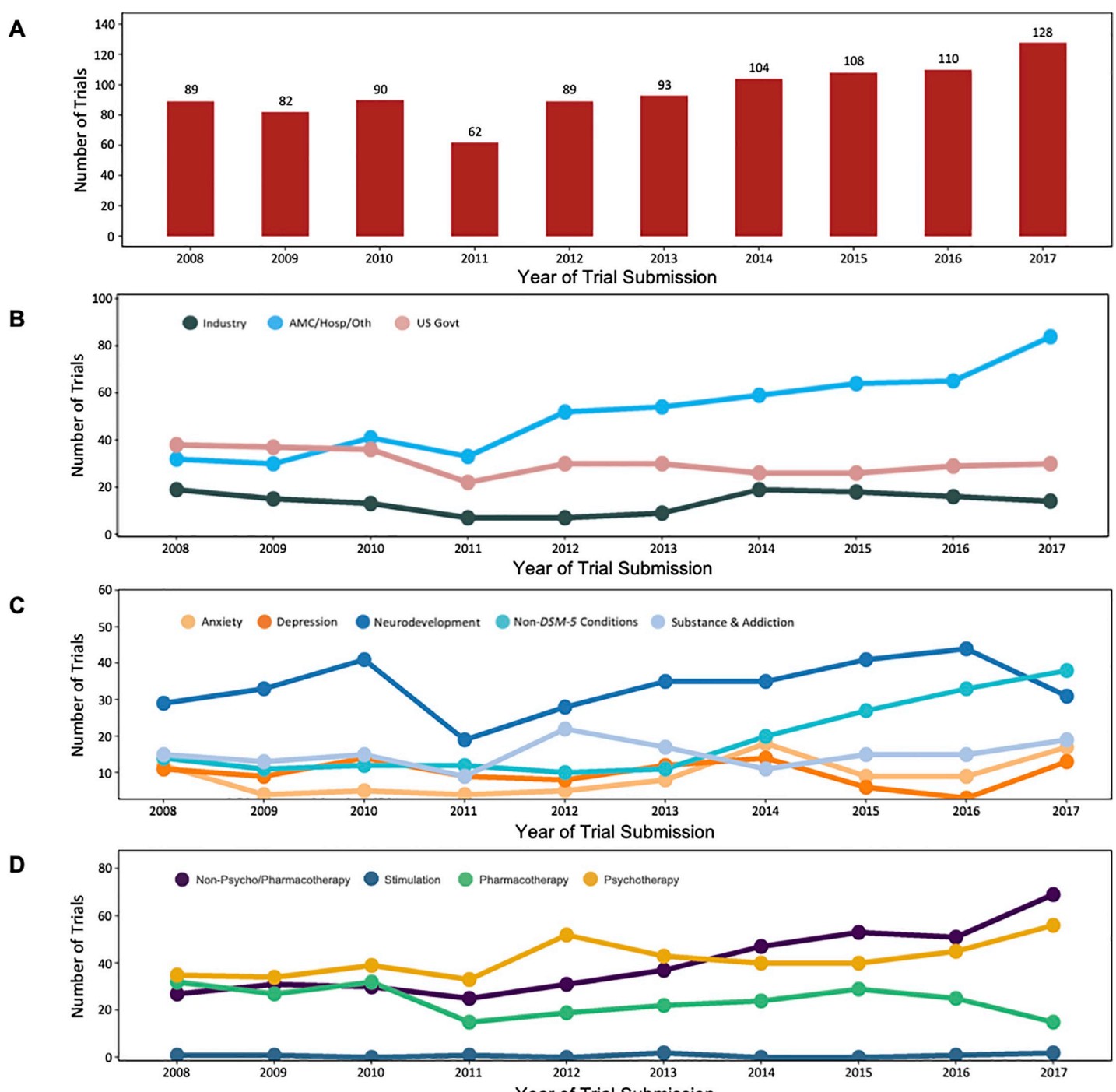

**Fig 2. Trends in the growth, funding, and disorders and treatments studied in US pediatric mental health trials registered in ClinicalTrials.gov from 2008 to 2017.** All year-to-year analyses included only years with a full 12-month collection of data (i.e., January 1, 2008 –December 31, 2017). (A) Overall growth of US pediatric mental health trials (CAGR 4.1%, p = 0.0030). (B) Growth of US pediatric mental health trials stratified by funder type. Industry CAGR -3.3%, p = 1.0; AMC/Hosp/Oth (Academic Medical Centers/Hospitals/Others) CAGR 11.3%, p = 0.00034; US Govt (US Government) CAGR -2.6%, p = 0.10. (C) Growth of the five most-studied disorder categories in US pediatric mental health. Anxiety (CAGR 3.9%, p = 0.085); Depression (CAGR 1.9%, p = 0.59); Neurodevelopment (CAGR 5.7%, p = 0.21); Non-*DSM-5* Conditions (CAGR 11.7%, p = 0.047); Substance & Addiction (CAGR 2.7%, p = 0.46). (D) Growth of US pediatric mental health trials stratified by treatment type. Non-Psycho/Pharmacotherapy (CAGR 11.0%, p = 0.00034); Psychotherapy (CAGR 5.4%, p = 0.039); Pharmacotherapy (CAGR -8.1%, p = 0.37); Stimulation (too few to calculate a meaningful CAGR).

**Table 1. Characteristics of US pediatric mental health clinical trials registered in ClinicalTrials.gov from October 1, 2007 to April 30, 2018 stratified by early (2007–2012) and late (2013–2018) time periods.**

| Trial Characteristics | Early | Late | Total | p-value |
|---|---|---|---|---|
| | n (%) | n (%) | n (%) | |
| **Primary Objective** | n = 432 | n = 573 | n = 1005 | |
| Treatment | 329 (76.2) | 356 (62.1) | 685 (68.2) | <0.0001 |
| Prevention | 71 (16.4) | 113 (19.7) | 184 (18.3) | 0.18 |
| Supportive Care | 8 (1.9) | 28 (4.9) | 36 (3.6) | 0.010 |
| Other | 24 (5.6) | 76 (13.3) | 100 (10.0) | <0.0001 |
| **Trial Phases** | n = 441 | n = 578 | n = 1019 | |
| Phase 1 | 42 (9.5) | 28 (4.8) | 70 (6.9) | 0.0034 |
| Phase 1/2-2 | 103 (23.4) | 60 (10.4) | 163 (16.0) | <0.0001 |
| Phase 2/3-3 | 49 (11.1) | 40 (6.9) | 89 (8.7) | 0.019 |
| Phase 4 | 53 (12.0) | 34 (5.9) | 87 (8.5) | 0.00052 |
| Not Applicable | 194 (44.0) | 416 (72.0) | 610 (59.9) | <0.0001 |
| **Number of Arms** | n = 428 | n = 576 | n = 1004 | |
| One | 67 (15.7) | 98 (17.0) | 165 (16.4) | 0.57 |
| Two | 284 (66.4) | 388 (67.4) | 672 (66.9) | 0.74 |
| ≥Three | 77 (18.0) | 90 (15.6) | 167 (16.6) | 0.32 |
| **Blinding** | n = 435 | n = 578 | n = 1013 | |
| Double | 114 (26.2) | 104 (18.0) | 218 (21.5) | 0.0016 |
| Single | 137 (31.5) | 198 (34.3) | 335 (33.1) | 0.36 |
| None | 184 (42.3) | 276 (47.8) | 460 (45.4) | 0.084 |
| **Randomization** | n = 436 | n = 577 | n = 1013 | |
| Yes | 349 (80.0) | 450 (78.0) | 799 (78.9) | 0.47 |
| **DMC** | n = 419 | n = 547 | n = 966 | |
| Yes | 203 (48.4) | 221 (40.4) | 424 (43.9) | 0.015 |
| **Number of Sites** | n = 441 | n = 578 | n = 1019 | |
| One | 336 (76.2) | 447 (77.3) | 783 (76.8) | 0.67 |
| Two | 37 (8.4) | 57 (9.9) | 94 (9.2) | 0.42 |
| Three-Ten | 40 (9.1) | 43 (7.4) | 83 (8.1) | 0.35 |
| >Ten | 28 (6.3) | 31 (5.4) | 59 (5.8) | 0.50 |
| **Funder** | n = 441 | n = 578 | n = 1019 | |
| Industry | 69 (15.6) | 79 (13.7) | 148 (14.5) | 0.37 |
| AMC/Hosp/Oth | 200 (45.4) | 342 (59.2) | 542 (53.2) | <0.0001 |
| US Govt | 172 (39.0) | 157 (27.2) | 329 (32.3) | <0.0001 |
| **Treatment Type** | n = 441 | n = 578 | n = 1019 | |
| Psychotherapy | 203 (46.0) | 236 (40.8) | 439 (43.1) | 0.11 |
| Pharmacotherapy | 140 (31.7) | 119 (20.6) | 259 (25.4) | <0.0001 |
| Stimulation | 4 (0.9) | 5 (0.9) | 9 (0.9) | - |
| Non-Psycho/Pharmacotherapy | 149 (33.8) | 283 (49.0) | 432 (42.4) | <0.0001 |
| **Disorder Category** | n = 441 | n = 578 | n = 1019 | |
| Anxiety | 33 (7.5) | 64 (11.1) | 97 (9.5) | 0.068 |
| Bipolar | 19 (4.3) | 24 (4.2) | 43 (4.2) | 1.0 |
| Depression | 54 (12.2) | 50 (8.7) | 104 (10.2) | 0.076 |
| Disruptive, Impulse Control, & Conduct | 15 (3.4) | 26 (4.5) | 41 (4.0) | 0.47 |
| Dissociative | 1 (0.2) | 0 | 1 (0.1) | - |
| Feeding & Eating | 12 (2.7) | 10 (1.7) | 22 (2.2) | - |
| Gender Dysphoria | 0 | 2 (0.3) | 2 (0.2) | - |

*(Continued)*

**Table 1.** (Continued)

| Trial Characteristics | Early | Late | Total | p-value |
|---|---|---|---|---|
| | n (%) | n (%) | n (%) | |
| Neurocognitive | 4 (0.9) | 7 (1.2) | 11 (1.1) | - |
| Neurodevelopment | 161 (36.5) | 192 (33.2) | 353 (34.6) | 0.30 |
| Obsessive-Compulsive | 17 (3.9) | 11 (1.9) | 28 (2.7) | - |
| Paraphilic | 0 | 1 (0.2) | 1 (0.1) | - |
| Personality | 0 | 0 | 0 | - |
| Schizophrenia Spectrum | 8 (1.8) | 8 (1.4) | 16 (1.6) | - |
| Sexual Dysfunction | 0 | 1 (0.2) | 1 (0.1) | - |
| Sleep-Wake | 9 (2.0) | 15 (2.6) | 24 (2.4) | - |
| Somatic Symptom | 1 (0.2) | 5 (0.9) | 6 (0.6) | - |
| Substance & Addiction | 78 (17.7) | 88 (15.2) | 166 (16.3) | 0.33 |
| Trauma & Stressor | 17 (3.9) | 21 (3.6) | 38 (3.7) | 0.99 |
| Non-*DSM-5* Condition | 64 (14.5) | 142 (24.6) | 206 (20.2) | 0.00010 |

'AMC/Hosp/Oth' denotes Academic Medical Centers/Hospitals/Other. 'US Govt' denotes United States Government. DMC denotes oversight by a data monitoring committee. Non-*DSM-5* conditions were disorders that did not clearly match any *Diagnostic Statistical Manual-5* disorder categories. Of note, the total number of trials varies slightly by category, as approximately 5% of trials had missing dimensions (n provided for each category). For the disorder and treatment categories, trials were labeled with as many categories as were relevant, and consequently the total percentage of trials by disorder and treatment categories sums to greater than 100%. For the 11 diagnostic categories that had fewer than 30 trials, we did not calculate $\chi^2$ values (represented as dashes). The same was true for the treatment 'Stimulation', which had fewer than 30 trials. All p-values are from two-sided Pearson $\chi^2$ tests.

p = 1.0), and the proportion of industry-funded trials did not change substantively (69 to 79 trials, 15.6% to 13.7%, p = 0.37). Trials funded through academic medical center/hospital/other sources grew monotonically (CAGR 11.3%, p = 0.00034) and proportionally (200 to 342 trials, 45.5% to 59.2%, p<0.0001).

Of the five most-studied disorders (Fig 2C; Table 1), only Non-*DSM-5* conditions trended towards monotonic growth (CAGR 11.7%, p = 0.047) and grew proportionally (14.5% to 24.6%, p = 0.00010). None of the other conditions grew monotonically (Anxiety CAGR 3.9%, p = 0.085; Depression CAGR 1.9%, p = 0.59; Neurodevelopment CAGR 5.7%, p = 0.21; Substance CAGR 2.7%, p = 0.46) or grew proportionally (all: p>0.005). Growth of pediatric mental health trials differed by treatment type as well (Fig 2D; Table 1). Trials studying non-psycho/pharmacotherapy treatments grew monotonically (CAGR 11.0%, p = 0.00034) and psychotherapy treatments trended towards growth (CAGR 5.4%, p = 0.039). The overall proportion of trials studying psychotherapy trended downward (46.0% to 40.8%, p = 0.11) while the proportion of trials studying non-psycho/pharmacotherapy treatments grew significantly (33.8% to 49.0%, p<0.0001). The proportion of trials studying pharmacotherapy declined between the early and late periods (31.7% to 20.6%, p<0.0001), though this trend was not monotonic (CAGR -8.1%, p = 0.37). Only nine stimulation trials were conducted during this time. There were too few trials to calculate a meaningful CAGR, and there was no proportional change.

There were multiple changes in trial design characteristics between the early and late time periods (Table 1). Trial objectives shifted away from treatment (76.2% to 62.1%, p<0.0001) and trended towards Supportive Care (1.9% to 4.9%, p = 0.010) and towards Other objectives (5.6% to 13.3%, p<0.0001). There was a significant increase in the proportion of trials with 'Not Applicable' phase designation (44.0% to 72.0%, p<0.0001), and there was a decline in trials with double blinding (26.2% to 18.0%; p = 0.0016) and trend towards decline of oversight

by a data monitoring committee (DMC, 48.4% to 40.4%; p = 0.015). There were no significant changes in the proportions of trials with multiple arms, the number of study sites, and trials using randomization (all: p>0.005).

## Disorders studied

The top five disorder categories studied (neurodevelopment, substance & addiction, depression, anxiety, and Non-*DSM-5* conditions) comprised 90.8% of pediatric mental health trials (Fig 3A). The remaining 14 disorder categories were studied in 23.0% of trials. Trials were labeled with as many disorder categories as were relevant, and consequently the total percentage of trials by disorder category sums to greater than 100%. There were marked differences in the proportions of treatment types studied in each disorder category (Fig 3B). For example, trials studying neurodevelopment (Pharm 40.3%, Psycho 28.8%, Non-Psycho/Pharm 30.4%), bipolar (Pharm 52.2%, Psycho 21.7%, Non-Psycho/Pharm 26.1%), and Obsessive-Compulsive (Pharm 40.0%, Psycho 48.6%, Non-Psycho/Pharm 11.4%) disorders studied relatively high proportions of pharmacotherapies. Conversely, trials studying disorder categories such as substance & addiction (Pharm 8.7%, Psycho 43.9%, Non-Psycho/Pharm 47.4%), depression (Pharm 12.0%, Psycho 44.4%, Non-Psycho/Pharm 37.6%), anxiety (Pharm 13.3%, Psycho 53.3%, Non-Psycho/Pharm 33.3%), disruptive, impulse control, & conduct (Pharm 4.1%, Psycho 53.1%, Non-Psycho/Pharm 42.9%), trauma & stressor disorders (Pharm 6.4%, Psycho 51.1%, Non-Psycho/Pharm 42.6%) and Non-*DSM-5* conditions (Pharm 9.4%, Psycho 33.2%, Non-Psycho/Pharm 54.3%) studied relatively few pharmacotherapies compared to psychotherapy and non-psycho/pharmacotherapy treatments. Stimulation trials comprised ≤6% of trials across all disorder categories.

There were marked differences in the proportions of disorder categories studied by each funding source (Fig 3C). Academic medical centers/hospitals/others funded the largest proportion of trials in almost all disorder categories. Industry funded the smallest proportion of trials in all disorder categories except for neurodevelopment (Ind 29.2%, AMC/Hosp/Oth 49.6%, US Govt 21.2%). The emphasis each funder placed on studying each disorder type also differed (Table 2). For example, industry devoted the majority of its trials to studying neurodevelopment (69.6%), which was a significantly larger proportion compared to the other funders (Ind 69.6%, AMC/Hosp/Oth 32.3%, US Govt 22.8%; p<0.0001). Academic medical centers/hospitals/others funded the largest proportion of trials studying Non-*DSM-5* conditions (Ind 8.1%, AMC/Hosp/Oth 23.4%, US Govt: 20.4%; p = 0.00021) and anxiety disorders (Ind 2.7%, AMC/Hosp/Oth 11.4%, US Govt 9.4%; p = 0.0058), and it devoted the largest proportion of its trials to neurodevelopment (32.3%). The US government devoted the largest proportion of trials to studying substance & addiction (Ind 4.7%, AMC/Hosp/Oth 14.4%, US Govt 24.6%; p<0.0001), and it devoted a large proportion of its trials to Non-*DSM-5* conditions (20.4%) and neurodevelopment (22.8%).

## Treatments studied

Fig 4A shows that non-psycho/pharmacotherapy and psychotherapy treatments were studied roughly equally (42.4% and 43.1%, respectively), pharmacotherapies were studied in 25.4% of trials, and stimulation treatments comprised only 0.9% of trials. Treatment types were further broken down into subcategories for non-psycho/pharmacotherapy, pharmacotherapy, and psychotherapy treatment categories (Fig 4B–4D). The largest proportion of non-psycho/pharmacotherapy treatments were community interventions (Community Programs: 21.1%, e.g., afterschool programs and substance use prevention campaigns; Community Outreach– 7.4%, e.g., assertive community treatment teams and integration of mental health services into

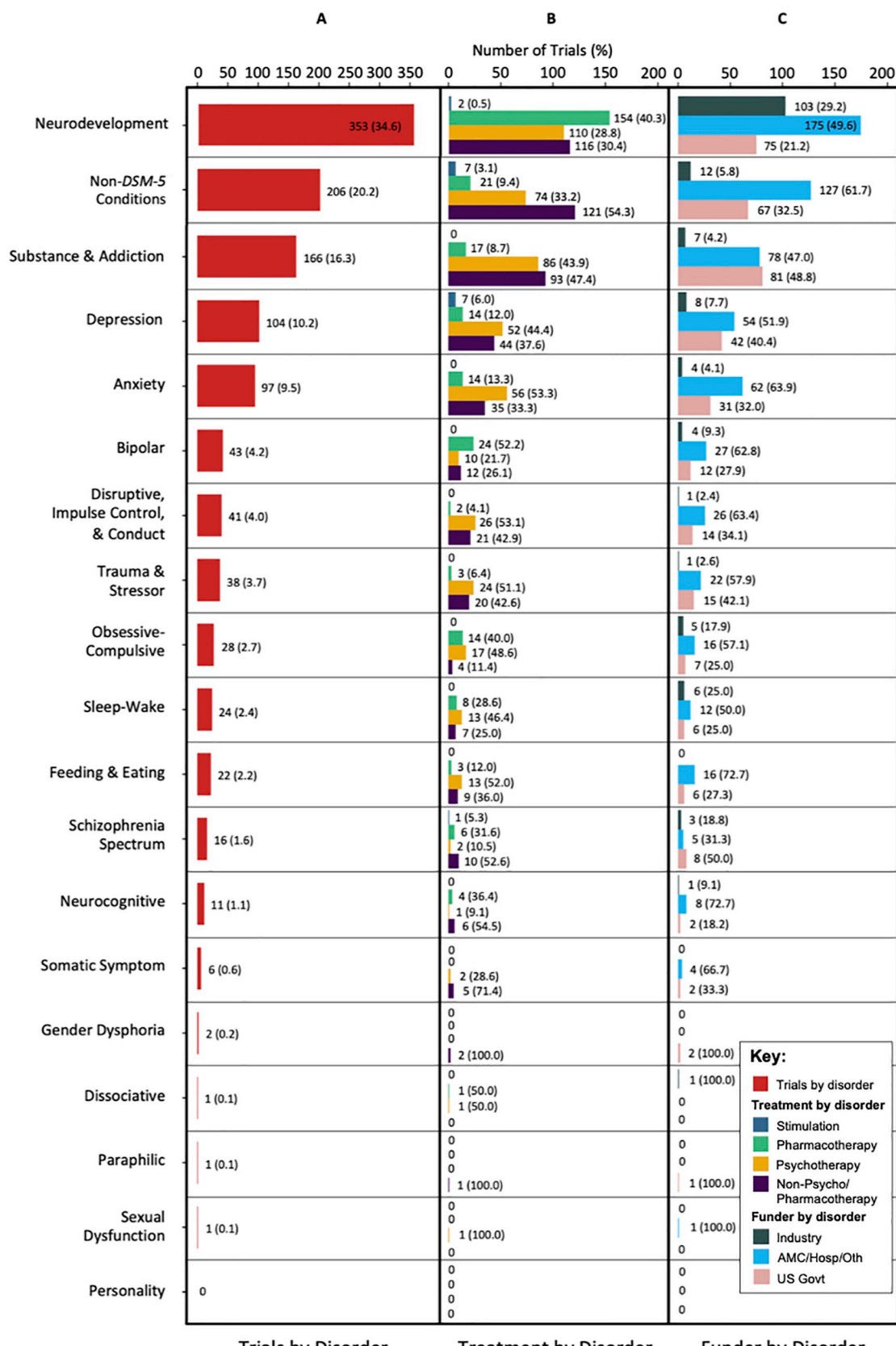

**Fig 3. US pediatric mental health trials registered in ClinicalTrials.gov from October 1, 2007 to April 30, 2018 stratified by disorder category.** (A) Number (percentage) of US pediatric mental health trials by disorder category. Trials were labeled with as many disorder categories as were relevant, and consequently the total percentage of trials by disorder category sums to greater than 100%. (B) Number (percentage) of treatments studied in US pediatric mental health trials by disorder category. Percentages were calculated for the proportion of treatments studied for each disorder category; therefore, each category sums to 100%. (C) Number (percentage) of funders of US pediatric mental health trials by disorder categories. Percentages were calculated for the proportion of funders for each disorder category; therefore, each category sums to 100%.

primary care) and technological interventions (Technology: 16.8%; e.g., interactive phone applications and video games; Telecommunication: 10.5%, e.g. telepsychiatry/teletherapy), which together comprised nearly two-thirds of the category (Fig 4B). Stimulants comprised the largest subcategory of pharmacotherapies studied (26.3%), followed by antipsychotics (14.3%) and antidepressants (12.7%). Educational and behavioral interventions comprised the majority of the psychotherapy interventions (59.7%), followed by cognitive behavioral therapy (23.7%).

**Table 2. Disorders and treatments studied in US pediatric mental health clinical trials registered in ClinicalTrials.gov from October 1, 2007 to April 30, 2018 stratified by funder type.**

| Trial Characteristics | Industry n (%) | AMC/Hosp/Oth n (%) | US Govt n (%) | p-value |
|---|---|---|---|---|
| **Disorder Category Studied** | n = 148 | n = 542 | n = 329 | |
| Anxiety | 4 (2.7) | 62 (11.4) | 31 (9.4) | 0.0058 |
| Bipolar | 4 (2.7) | 27 (5.0) | 12 (3.6) | 0.39 |
| Depression | 8 (5.4) | 54 (10.0) | 42 (12.8) | 0.047 |
| Disruptive, Impulse Control, & Conduct | 1 (0.7) | 26 (4.8) | 14 (4.3) | 0.075 |
| Dissociative | 1 (0.7) | 0 | 0 | - |
| Feeding & Eating | 0 | 16 (3.0) | 6 (1.8) | - |
| Gender Dysphoria | 0 | 0 | 2 (0.6) | - |
| Neurocognitive | 1 (0.7) | 8 (1.5) | 2 (0.6) | - |
| Neurodevelopment | 103 (69.6) | 175 (32.3) | 75 (22.8) | <0.0001 |
| Obsessive-Compulsive | 5 (3.4) | 16 (3.0) | 7 (2.1) | - |
| Paraphilic | 0 | 0 | 1 (0.3) | - |
| Personality | 0 | 0 | 0 | - |
| Schizophrenia Spectrum | 3 (2.0) | 5 (0.9) | 8 (2.4) | - |
| Sexual Dysfunction | 0 | 1 (0.2) | 0 | - |
| Sleep-Wake | 6 (4.1) | 12 (2.2) | 6 (1.8) | - |
| Somatic Symptom | 0 | 4 (0.7) | 2 (0.6) | - |
| Substance & Addiction | 7 (4.7) | 78 (14.4) | 81 (24.6) | <0.0001 |
| Trauma & Stressor | 1 (0.7) | 22 (4.1) | 15 (4.6) | 0.098 |
| Non-*DSM-5* Conditions | 12 (8.1) | 127 (23.4) | 67 (20.4) | 0.00021 |
| **Treatment Type Studied** | n = 148 | n = 542 | n = 329 | |
| Psychotherapy | 12 (8.1) | 255 (47.0) | 172 (52.3) | <0.0001 |
| Pharmacotherapy | 115 (77.7) | 96 (17.7) | 48 (14.6) | <0.0001 |
| Stimulation | 1 (0.7) | 5 (0.9) | 3 (0.9) | - |
| Non-Psycho/Pharmacotherapy | 34 (23.0) | 246 (45.4) | 152 (46.2) | <0.0001 |

'AMC/Hosp/Oth' denotes Academic Medical Centers/Hospitals/Other. 'US Govt' denotes United States Government. Non-*DSM-5* conditions were disorders that did not clearly match any *Diagnostic Statistical Manual-5* disorder categories. Trials were labeled with as many categories as were relevant, and consequently the total percentage of trials by disorder and treatment categories sums to greater than 100%. For the 11 diagnostic categories that had fewer than 30 trials, we did not calculate $\chi^2$ values (represented as dashes). The same was true for the treatment 'Stimulation', which had fewer than 30 trials. All p-values are from two-sided Pearson $\chi^2$ tests.

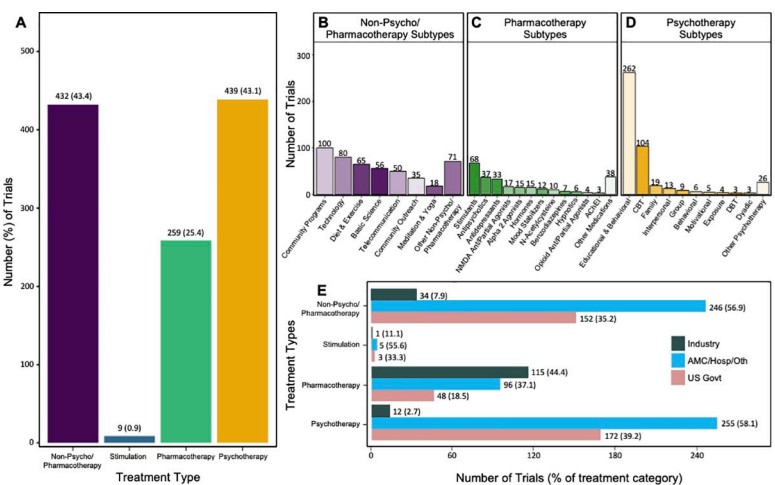

**Fig 4. US pediatric mental health trials registered in ClinicalTrials.gov from October 1, 2007 to April 30, 2018 stratified by treatment type.** (A) US pediatric mental health trials stratified by treatment type. Trials were labeled with as many treatment categories as were relevant, and consequently the total percentage of trials by treatment category sums to greater than 100%. (B-D) Treatment categories (i.e., non-psycho/pharmacotherapy, pharmacotherapy, and psychotherapy) for US pediatric mental health trials stratified into subtypes. Trials were labeled with as many treatment subtypes as were relevant, and consequently the total percentage of trials by subcategory sums to greater than 100%. Acronyms: AChEI (acetylcholinesterase inhibitors), CBT (cognitive behavioral therapy), DBT (dialectical behavioral therapy). (E) Treatment types studied in US pediatric mental health trials stratified by funder type. Percentages were calculated for the proportion of funders for each treatment category; therefore, each category sums to 100%.

The types of treatments studied also differed by funding source (Fig 4E). Overall, industry funded the largest percentage of trials studying pharmacotherapies (Ind 44.4%, AMC/Hosp/Oth 37.1%, US Govt 18.5%), and academic medical centers/hospitals/others funded the largest percentage of trials studying psychotherapy (Ind 2.7%, AMC/Hosp/Oth 58.1%, US Govt 39.2%) and non-psycho/pharmacotherapy treatments (Ind 7.9%, AMC/Hosp/Oth 56.9%, US Govt 35.2%). There were too few stimulation trials to make a meaningful comparison of funder types. The emphasis each funder placed on studying each treatment type also differed (Table 2). Industry devoted a substantially larger proportion of its trials to studying pharmacotherapy compared to the other funders (Ind 77.7%, AMC/Hosp/Oth 17.7%, US Govt 14.6%; p<0.0001). Conversely, academic medical center/hospital/other and US government funders devoted a larger proportion of their trials to psychotherapies (Ind 8.1%, AMC/Hosp/Oth 47.0%, US Govt 52.3%; p<0.0001) and non-psycho/pharmacotherapy treatments (Ind 23.0%, AMC/Hosp/Oth 45.5%, US Govt 46.2%; p<0.0001) compared to industry (Table 2). Stimulation trials comprised a similar proportion of all three funder types (Ind 0.7%, AMC/Hosp/Oth 0.9%, US Govt 0.9%).

## Discussion

This study described the landscape, and changes, in contemporary US pediatric mental health trials in the ClinicalTrials.gov registry over the past decade. There were multiple primary findings. US pediatric mental health trials grew over the past decade. The number of academic medical center/hospital/other funded trials grew, while the number of industry and US government-funded trials remained unchanged. Neurodevelopmental disorders comprised the largest proportion of disorders studied; trials studying Non-*DSM-5* conditions comprised the only disorder category to grow. The disorders studied differed by funding source. There was significant growth of trials studying non-psycho/pharmacotherapy treatments during this

time period, with proportional decline of trials studying pharmacotherapies. Trial characteristics also changed, with a decline in trials using double blinding.

From 2007 to 2018, pediatric mental health trials grew (CAGR 4.1%) at approximately twice the rate of all mental health clinical research (CAGR 2.2%) [17]. Growth of both may have been driven by increasing numbers of academic medical centers and hospitals pursuing philanthropic support [33], but the slower growth of all mental health research can be attributed to a general decline in industry and US government funding that did not occur in trials studying pediatric populations [17]. There are several possible explanations for this preserved US government and industry funding for pediatric mental health research. Despite a 42% decrease in the NIMH total budget from 2005 to 2015 [34], the US government started the Autism Center of Excellence and devoted new funds to researching eating disorders and substance use disorders during this time [35, 36]. The government's emphasis on studying pediatric mental health disorders impacted priorities in industry as well by creating incentives for industry to develop medications for children [37]. Most recently there have been a number of industry-funded pediatric mental health trials studying long-acting stimulants to treat ADHD [38]. This has occurred while industry has significantly divested from researching psychotropic agents to treat disorders such as depression, bipolar disorder, and schizophrenia in adult populations [39].

The preponderance of pediatric mental health trials studying neurodevelopmental disorders (Total 34.6%; Ind 69.6%, AMC/Hosp/Oth 32.3%, US Govt 22.8%) may reflect a response to rising rates of diagnosing autism and ADHD in children [6, 8]; however, it may also be a reaction to industry's development of new stimulants, which we identified comprised 26.3% of all pediatric medication trials during this time period. Between 2014 and 2018, industry spent over $11 million marketing stimulants to psychiatrists [40]. This coincided with a doubling of stimulant prescriptions from 2006 to 2016 [41], and as of 2013, stimulants comprised the largest grossing class of medications for children [42]. It is worth considering how much of our focus on researching neurodevelopmental disorders is driven by patient need versus market forces. For example, 0.5% to 3.0% of children have obsessive-compulsive disorder (OCD) [43], which is at least as prevalent as autism, yet trials studying OCD comprised only 2.7% of our sample.

Non-*DSM-5* conditions comprised the only disorder category to show proportional growth (14.5% to 24.6%) over the past decade. This mirrors what was observed across all mental health clinical research from 2007 to 2018 [17]. One possible explanation for this trend is the growing adoption of the NIMH Research Domain Criteria (RDoC) initiative, which was started in 2009 but only became a component of NIMH grant reviews starting in 2012 to 2013 [44]. This coincides with the observed growth of trials studying Non-*DSM-5* conditions starting in 2014 (Fig 2C). RDoC has been an effort by the NIMH to move away from studying categorical, *DSM* diagnoses and towards studying dimensional brain systems and endophenotypes, which often cross traditional diagnostic boundaries [45, 46]. RDoC's focus on exploring neural networks and basic biology may also help explain the trend for pediatric mental health trials to focus less on treatment and more on other primary goals, including basic science [47]. If RDoC is indeed causing this growth of Non-*DSM-5* conditions, there is evidence to suggest that RDoC has had a significant impact on the characterization of US pediatric mental health diagnoses in clinical research as of 2014.

Another notable trend over the past decade has been growth in the number of pediatric mental health trials studying non-psycho/pharmacotherapy treatments (CAGR 11.0%), as well as the trend towards studying more psychotherapy treatments (CAGR 5.4%). These changes were also observed across all mental health trials during this time period [17]. Further parsing of these two treatment categories shows that the majority of pediatric mental health trials

studied community-based, technology, or educational/behavioral interventions (Fig 4B–4D). This emphasis on both non-psycho/pharmacotherapy and psychotherapy treatments may reflect research priorities of the US government and US surgeon general, who have identified "natural settings," particularly schools, as the most effective places to provide mental health treatment and preventative services to children and adolescents [48, 49]. This is because school-based mental health providers can observe patients in their natural milieu, have easier access to teachers for collateral and psychoeducation, are more accessible to children with limited transportation or support from family, and are felt to be less stigmatizing [50]. Funding for school and community-based trials have received significant support on the state level. For example, from 2007–2010, the Minnesota Department of Human Services alone apportioned over $10 million to develop the infrastructure for school-based mental health services [51]. For similar reasons, there has been increasing investment in studies of technological interventions, such as phone apps for telepsychiatry and teletherapy, to facilitate access to mental health care for adolescents in the community [52]. Community outreach programs and technological interventions have proven essential for treating mental health disorders during the COVID-19 global pandemic [53], and these modalities will likely continue growing in importance and prevalence in pediatric mental health clinical research.

The decline in trials using double blinding (26.2% to 18.0%) and the trend towards a decline in use of DMCs (48.4% to 40.4%) likely has to do with the proportional increase in trials studying non-psycho/pharmacotherapy therapies. It is often not possible to double- or even single-blind these interventions, and US regulations only require DMC oversight in trials testing new drugs/biologics/devices, in double-blinded studies with considerable risk to patients, or in research with vulnerable populations [54]. A decrease in double-blinded trials and use of DMCs has also been observed in the registry across all mental health clinical research [17].

It is interesting to consider why there are differences among disorder categories regarding the proportions of pharmacotherapy vs non-pharmacotherapy treatments studied in children. For example, in certain disorder categories, such as neurodevelopment, a significantly larger proportion of pharmacotherapies were studied (Pharm 40.3% vs Psycho 28.8% and Non-Psycho/Pharm 30.4%), whereas in other disorders, such as substance & addiction, a larger proportion of trials studied psychotherapy and non-psycho/pharmacotherapy treatments (Pharm 8.7% vs Psycho 43.9% and Non-Psycho/Pharm 47.4%). This may reflect differences in the accepted effectiveness of certain treatments for different disorders. It is also possible that patient and parent preferences may be driving a shift towards studying psychotherapies for certain conditions. Analyses across diverse clinical settings show that mental health patients, particularly younger patients, express a three-fold preference for psychotherapies over medications [55]. This may be especially true for disorders in which pharmacotherapy and psychotherapy have equal effectiveness [56, 57]. It is also interesting to consider why so few pediatric mental health trials studied stimulation therapies. Pilot studies using transcranial magnetic stimulation to treat ADHD, autism, and depression in children have been promising, and it is likely that there will be growth of trials studying this treatment modality in children over the coming decades [58].

Because ClinicalTrials.gov is a unique and valuable resource to study trends in clinical research, it is important to consider ways it could be modified to improve its usefulness and efficiency as a research tool. For example, it would be helpful if trials reported the monetary contributions of all funding sources, as this would help establish each funder's relative contribution and influence on trial design and focus. It would be beneficial for trials to identify relevant fields of medicine (e.g., mental health, oncology, cardiology, etc.), as currently this information needs to be gleaned through use of MeSH and Disease Condition terms and manual review of the study titles and descriptions. This process is time consuming, has the

potential for error, and contributes to variation among study results. Lastly, it would be helpful for ClinicalTrials.gov to require all data fields to be mandatory, as missing data can potentially introduce bias into analyses of the registry.

Our study has several possible limitations. First, ClinicalTrials.gov is not an exhaustive list of all US clinical trials [13]. Phase 1 trials and trials studying non-pharmacologic interventions were not subject to the FDAAA or the Final Rule, so these trials may be underrepresented in the registry [9]. There may be other unknown norms and incentives that also bias the registration of certain trial types. Therefore, trends identified in the registry may at least in part reflect changes in trial registration rather than changes in clinical research. Analyses of the registry are also descriptive in nature and include many comparisons that do not account for potential unknown confounders. As a result, causal relationships cannot be drawn from these data. Nevertheless, analyses of ClinicalTrials.gov have allowed many medical specialties to assess trends in clinical research that might otherwise remain unassessable [12–14, 16, 17], as nearly half of trials run by large sponsors go unpublished [59]. Second, while significant efforts were made to review all key words, titles, and study descriptions to confirm trials' relevance to mental health, some trials may have been excluded due to missing or mislabeled keywords in the registry. Finally, we looked exclusively at US trials registered in ClinicalTrials.gov. International regulations for trial registration differ by country and were thought to likely confound trends if non-US trials were included in our sample. Consequently, 37.3% of the pediatric mental health trials registered in ClinicalTrials.gov from October 1, 2007 to April 30, 2018 were excluded, and our results cannot be generalized beyond the United States.

In conclusion, this study aims to help provide a mirror to the pediatric mental health community to identify where its clinical research efforts have been and where its efforts appear to be heading. By observing these trends, researchers and funding bodies may gain an additional perspective to help shape the priorities and resources devoted to future pediatric mental health research to provide new treatments to better meet patients' needs.

## Supporting information

**S1 Table. Changes to the initial protocol.**
(DOCX)

**S2 Table. Non-*DSM-5* subcategorization.**
(DOCX)

**S1 File. Study protocol.**
(DOCX)

## Author Contributions

**Conceptualization:** Joshua R. Wortzel, Brandon E. Turner, Brannon T. Weeks, Thomas G. O'Connor.

**Data curation:** Joshua R. Wortzel, Brandon E. Turner, Brannon T. Weeks.

**Formal analysis:** Joshua R. Wortzel, Brandon E. Turner, Brannon T. Weeks.

**Investigation:** Joshua R. Wortzel, Brandon E. Turner, Christopher Fragassi, Virginia Ramos, Thanh Truong, Desiree Li, Omar Sahak, Thomas G. O'Connor.

**Methodology:** Joshua R. Wortzel, Brandon E. Turner, Brannon T. Weeks, Christopher Fragassi, Virginia Ramos, Thanh Truong, Desiree Li, Omar Sahak, Thomas G. O'Connor.

**Project administration:** Joshua R. Wortzel.

**Supervision:** Thomas G. O'Connor.

**Validation:** Joshua R. Wortzel, Christopher Fragassi, Virginia Ramos, Thanh Truong, Desiree Li, Omar Sahak.

**Visualization:** Joshua R. Wortzel, Brandon E. Turner.

**Writing – original draft:** Joshua R. Wortzel, Brandon E. Turner, Thomas G. O'Connor.

**Writing – review & editing:** Joshua R. Wortzel, Brandon E. Turner, Christopher Fragassi, Virginia Ramos, Thanh Truong, Desiree Li, Omar Sahak, Thomas G. O'Connor.

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
