## [Decision Letter · Decision Letter 0]

5 Jan 2021

PONE-D-20-34754

Trends in US pediatric mental health clinical trials: An analysis of ClinicalTrials.gov from 2007 – 2018

PLOS ONE

Dear Dr. Wortzel,

Thank you for submitting your manuscript to PLOS ONE. After careful consideration, we feel that it has merit but does not fully meet PLOS ONE’s publication criteria as it currently stands. Therefore, we invite you to submit a revised version of the manuscript that addresses the points raised during the review process.

We look forward to receiving your revised manuscript.

Kind regards,

Claudio Gentili

Academic Editor

PLOS ONE

Journal Requirements:

2.We note that you have indicated that data from this study are available upon request. PLOS only allows data to be available upon request if there are legal or ethical restrictions on sharing data publicly. For information on unacceptable data access restrictions, please see http://journals.plos.org/plosone/s/data-availability#loc-unacceptable-data-access-restrictions.

Additional Editor Comments (if provided):

Please consider carefully the issues raised by the two reviewers. Particularly, both highlighted that a the moment the manuscritpt did not fullfil the PLOS one policy for data sharing. I strongly invite the authors to fulfil the policy requirements submitting the revised version of the manuscript

Reviewers' comments:

Reviewer's Responses to Questions

**Comments to the Author**

1. Is the manuscript technically sound, and do the data support the conclusions?

Reviewer #1: Partly

Reviewer #2: Yes

2. Has the statistical analysis been performed appropriately and rigorously? 

Reviewer #1: Yes

Reviewer #2: Yes

3. Have the authors made all data underlying the findings in their manuscript fully available?

Reviewer #1: No

Reviewer #2: No

4. Is the manuscript presented in an intelligible fashion and written in standard English?

Reviewer #1: Yes

Reviewer #2: Yes

5. Review Comments to the Author

Reviewer #1: Florian Naudet, MD, PhD, Rennes 1 University

Sorry for the delay in answering, mostly due to my clinical activity.

This is a very interesting paper. I have the following remarks.

Abstract:

- add some limitations to avoid any spin;

Introduction:

- add a systematic overview of overlapping projects on adults/child, including in other countries (e.g. a table summarizing evidence prior this study);

Methods:

- was there a pre-registration? If, no please make it explicit.

- Please state in the text the date of the protocol / the date of the analysis / and describe any change to the protocol in a dedicated paragraph;

- Please provide more detail about the supportive care category in the method section (and make sure that one can understand how it differs from treatment);

- How were managed missing data on clinicaltrials.gov (e.g. related to study phase)? Please add some details?

- When it comes to interventions tested, please replace the category “interventional” by another word (e.g. “stimulation”) … “Interventional” in my opinion rather refers to an interventional study (versus observational studies) and it could be somewhat misleading (e.g. it is used with this meaning page 5 line 73);

- No reason is given for the threshold of alpha=0.01. This would apply in case there are 5 primary outcomes following a Bonferroni correction. It must be justified more adequately.

In addition, and, to be provocative, I’m not sure that most of these p-values are needed (as you are describing the complete population, statistical testing does not really make sense in my view).

- We need somewhere the details of the “non-DSM” category (in addition, it is the most represented category at some points, e.g. the last years, a list in the methods or a figure in the results may be helpful);

Results:

- Please add a flow charts and a paragraph about study selection;

- Please detail the interrater agreement for data extraction;

- Table 1: see my comments about statistical testing.

- Please clarify this sentence as the numbers (90.8+23) do not add up to 100 %. I understand that there can be some overlap but it should be explained: “The top five disorder categories studied (neurodevelopment, substance & addiction, depression, anxiety, and Non-DSM-5 conditions) comprised 90.8% of pediatric mental health trials (Fig 2A). The remaining 14 disorder categories were studied in 23.0% of trials.”

- Figure 2 is really nice. Congratulation for this figure. Please consider moving “non-DSM” Disorders to the second line (and to order the graph by frequencies) ;

- Figure 3 is also a really nice one. For panels B, C and D, I would suggest however to use the same scale for the y-axis. I understand that the number are very different from one category to the other. So perhaps, you can use a log scale. Second possibility, you can use percentages (the numbers being given in panel A).

Discussion:

- Please insist on the descriptive nature of the study.

- As there are many comparisons without any adjustment of confounder. Please warn explicitly about the fact that no causal interpretations are possible, discuss the issue of confounding toroughly.

Overall:

I acknowledge that there are no appropriate reporting guidelines for such meta-research study. But please have a look at equator network and follow the most appropriate guideline (e.g. STROBE?).

Please share the data, code and any other information in an adequate repository (e.g. Dryad, https://datadryad.org/stash).

Reviewer #2: Major issues:

Data availability:

The authors should make the dataset used for the analysis available, it is not enough to just say where and how it could be retrieved. The authors might have used criteria or filters or other operations on a large dataset. Readers need to be able to replicate the main analyses. Is there a reason why the authors have opted not to share the extracted data?

How does this dataset differ the authors’ previous publication on all mental health research in the same journal (https://journals.plos.org/plosone/article?id=10.1371/journal.pone.0233996)? Weren’t pediatric trials a subset of that? What is the rationale of presenting a separate analysis on that cohort? At least a brief discussion of overlap and the rationale of the more detailed analysis on pediatric trials should be included.

Is there any reason for not pre-registering the study protocol? Submission with manuscript, while useful, does not afford the opportunity to assess what was pre-planned and what not.

My main objection is about treatment type categories, which appear rather vague and counter-intuitive, mixing type of intervention with delivery type or setting. The authors could consider reorganizing them. For instance, aren’t psychotherapy and pharmacotherapy also interventional? Also “alternative” is an unfortunate label that has a different meaning for many people, as in “alternative” to “conventional” biomedicine. I suggest you replace “Interventional” with “Physical”, as these treatments are often denominated. The “alternative” category is rather heterogenous, mixing types of interventions with modes of delivery or settings. I am also not sure how subcategories fit together, for example there are legitimate, evidence-based technology psychotherapies, whereas yoga or diet have been less studied. Moreover, it also overlaps with psychotherapy, often delivered via the Internet or in other electronic ways. Finally, where do prevention interventions enter?

What is the justification for the alpha threshold? Either the authors employ a correction for multiple comparisons, which is preferable, or at least use the most stringent threshold proposed (p=0.005, see https://www.nature.com/articles/s41562-017-0189-z and associated discussion).

I am also not sure all the comparisons in Table 2 make sense or have any relevance, particularly for categories with few trials. It might be more useful to just include statistical comparisons for a few pre-selected categories where there is a reasonable number of included trials, and just present the others descriptively.

Please list the relevant R packages used and also consider sharing the code for the analysis, to ensure reproducibility of findings.

Minor issues:

In the introduction, some discussion needs to be included about the fact that increases in prevalence might reflect changes in assessment criteria and instruments.

Please replace references 44 and 45 with more updated and comprehensive meta-analyses about the similar effectiveness of psychotherapy and pharmacotherapy. There are recent network meta-analyses, individual patient data meta-analyses and so on.

6. PLOS authors have the option to publish the peer review history of their article (what does this mean?). If published, this will include your full peer review and any attached files.

Reviewer #1: **Yes: **FLORIAN NAUDET

Reviewer #2: **Yes: **Ioana A. Cristea

---

## [Author Response · Author response to Decision Letter 0]

17 Feb 2021

Editor/reviewer comments are noted with an "**" before the comment. The responses are in the paragraph(s) below each comment.

Editor’s Comments:

**Comment 1: Please ensure that your manuscript meets PLOS ONE's style requirements, including those for file naming. The PLOS ONE style templates can be found at, https://journals.plos.org/plosone/s/file?id=wjVg/PLOSOne_formatting_sample_main_body.pdf and https://journals.plos.org/plosone/s/file?id=ba62/PLOSOne_formatting_sample_title_authors_affiliations.pdf.

Thank you for your guidance with this. We have reviewed these documents and believe that our manuscript now meets the PLOS ONE style and formatting requirements. 

**Comment 2: We note that you have indicated that data from this study are available upon request. PLOS only allows data to be available upon request if there are legal or ethical restrictions on sharing data publicly. For information on unacceptable data access restrictions, please see http://journals.plos.org/plosone/s/data-availability#loc-unacceptable-data-access-restrictions.

Thank you. We have downloaded our data and code onto a repository, and they are now freely available.

**Comment 3: In your revised cover letter, please address the following prompts: a) If there are ethical or legal restrictions on sharing a de-identified data set, please explain them in detail (e.g., data contain potentially identifying or sensitive patient information) and who has imposed them (e.g., an ethics committee). Please also provide contact information for a data access committee, ethics committee, or other institutional body to which data requests may be sent. b) If there are no restrictions, please upload the minimal anonymized data set necessary to replicate your study findings as either Supporting Information files or to a stable, public repository and provide us with the relevant URLs, DOIs, or accession numbers. Please see http://www.bmj.com/content/340/bmj.c181.long for guidelines on how to de-identify and prepare clinical data for publication. For a list of acceptable repositories, please see http://journals.plos.org/plosone/s/data-availability#loc-recommended-repositories. We will update your Data Availability statement on your behalf to reflect the information you provide.

We have now made our data and code freely available on an online repository. There are no restrictions to access this deidentified data. We have included the URL below as well as in the revised cover letter: https://datadryad.org/stash/share/8riF8rerFkmSGssL3DaMv4qb07I9W7CC4m3w387zseE. Of note, the DOI (10.5061/dryad.t4b8gtj1d) is currently being curated, but it should be made available shortly).

**Comment 4: Please include captions for your Supporting Information files at the end of your manuscript, and update any in-text citations to match accordingly. Please see our Supporting Information guidelines for more information: http://journals.plos.org/plosone/s/supporting-information.

Thank you for your guidance with this. We have placed our two supplemental tables with table titles and legends at the end of the manuscript under a new section labeled ‘Supporting Information’. 

Reviewer 1’s Comments:

** Comment 1: (Abstract) Add some limitations to avoid any spin.

Thank you for this feedback. We added two sentences to the abstract to address potential limitations (Tracked version, Lines 23-27).

**Comment 2: (Introduction) Add a systematic overview of overlapping projects on adults/child, including in other countries (e.g., a table summarizing evidence prior this study).

We appreciated this recommendation. In addition to the paragraph that summarizes overlapping projects in adult populations (Tracked version, Lines 52-64), we added a paragraph to the introduction that itemizes the systematic reviews and meta-analyses we identified that addressed trends in pediatric mental health (Tracked version, Lines 66-84). These studies were limited to reporting trends in the investigation of particular treatment types. We did not find reviews on larger, systemic trends in pediatric mental health research across diagnostic categories or across treatment types, and so we have not provided a table. In this paragraph we also added a brief discussion of why trends drawn exclusively from published research may provide a biased perspective on what clinical research is being conducted. A prior analysis of ClinicalTrials.gov showed that nearly 50% of pediatric trials do not reach publication within 4.5 years of study completion [1]. We hope the addition of this information to the introduction adds helpful background that identifies where gaps remain in identifying trends in pediatric mental health clinical research and the need for the analysis described in this study.

**Comment 3: (Methods) Was there a pre-registration? If, no please make it explicit.

This study was not pre-registered. We added a sentence stating this explicitly in the Materials and Methods section (Tracked version, Line 120).

**Comment 4: (Methods) Please state in the text the date of the protocol / the date of the analysis / and describe any change to the protocol in a dedicated paragraph.

Thank you for your guidance with this. We have added two paragraphs to the Materials and Methods section detailing the 6 changes that were made to the original protocol, why they were made, the dates of when these changes were made, and when the analysis was conducted (Tracked version, Lines 119-153). We have also created a supplemental table (S1 Table) that summarizes these changes to the protocol, and we have uploaded a version of the protocol in which these changes have been tracked. You will see that four of the changes were due to recommendations we received from reviewers regarding our first manuscript (Wortzel et al., 2020) that were relevant to this analysis as well [2]. The most notable of these was confining our analysis to US trials, which removed a significant potential source of bias from the analysis. The other two changes to the protocol were made in response to reviewer comments for this manuscript. These included changing the alpha threshold for statistical significance to 0.005 (please see our response to your Comment 8 for an explanation of this change), and we changed the name of two of the intervention categories (‘Interventional’ to ‘Stimulation’ and ‘Alternative’ to ‘Non-Psycho/Pharmacotherapy’). We hope our additions help clarify these issues. 

**Comment 5: (Methods) Please provide more detail about the supportive care category in the method section (and make sure that one can understand how it differs from treatment).

Thank you for this suggestion. We have added a description of the ‘Supportive Care’ category. We also added definitions of the ‘Treatment’ and ‘Prevention’ categories to provide clarity of what constituted these trials’ ‘Primary Objectives’ (Tracked version, Lines 168-173).

**Comment 6: (Methods) How were managed missing data on clinicaltrials.gov (e.g., related to study phase)? Please add some details?

We appreciated this feedback. We added three sentences to address this issue. First, we added a sentence to clarify that the ‘N/A’ designation for ‘Study Phase’ does not refer to missing data but to trials that did not meet criteria for study phase designation (i.e., this is a category within ClinicalTrials.gov that primary investigators could choose if their trials did not have FDA-defined phases, such as trials studying devices or behavioral interventions) (Tracked version, Lines 179-181). We also added two sentences clarifying that, when data were missing for certain variables, trials with these missing data were excluded (Tracked version, Lines 242-244). The sample size used to assess each trial characteristic is reported in Table 1 to clarify when the sample size differed for each variable due to missing data. 

**Comment 7: (Methods) When it comes to interventions tested, please replace the category “interventional” by another word (e.g., “stimulation”) … “Interventional” in my opinion rather refers to an interventional study (versus observational studies) and it could be somewhat misleading (e.g., it is used with this meaning page 5 line 73).

Thank you – we agree – we appreciate how the term ‘Interventional’ can be confusing. We have changed the category ‘Interventional’ to ‘Stimulation’ as you suggested throughout the manuscript. 

**Comment 8: (Methods) No reason is given for the threshold of alpha=0.01. This would apply in case there are 5 primary outcomes following a Bonferroni correction. It must be justified more adequately. In addition, and, to be provocative, I’m not sure that most of these p-values are needed (as you are describing the complete population, statistical testing does not really make sense in my view).

Thank you for your guidance with this. This was also feedback that we received from Reviewer 2. Reviewer 2 recommended that we use a more stringent alpha=0.005, as has been previously published [3]. We appreciate your point that these are data describing the complete population of mental health trials registered in ClinicalTrials.gov, and we considered not calculating p-values for this reason. However, given that ClinicalTrials.gov likely does not contain 100% of the mental health trials conducted in the US (i.e., there are other registries in which trials could be registered, and technically not all types of trials are required to register [4]), we thought it was best to still consider these data as a sample of the entire pediatric mental health clinical trials portfolio, and to, therefore, report findings with p-values. Throughout the manuscript we also report information on percentage changes, which provide context for the effect sizes for the analyses.

**Comment 9: (Methods) We need somewhere the details of the “non-DSM” category (in addition, it is the most represented category at some points, e.g., the last years, a list in the methods or a figure in the results may be helpful).

We greatly appreciated your suggestion to delve into the ‘Non-DSM-5’ category further. We think that this adds clarity to a what is being studied in this category. We re-reviewed the titles and trial descriptions for these 206 trials, and we assigned them to 9 subcategories of ailments that are not defined in the DSM-5. We have created a supplemental table (S2 Table) that shows the breakdown of ‘Non-DSM-5’ trials within these 9 subcategories). 

**Comment 10: (Results) Please add a flow chart and a paragraph about study selection.

Thank you for this suggestion. We have created a flowchart showing how trials were selected, and we added this figure to the Results section. The origins of the number of trials used to calculate the ratios of US to global pediatric mental health trials (1,019/1,626) and pediatric to total US mental health trials (1,019/6,302) was previously unclear, and we think this flowchart provides needed clarity. We also dedicated a new paragraph (it appears as two paragraphs in the tracked revisions) in the results section to describe how trials were selected for inclusion in this analysis (Tracked version, Lines 260-270).

**Comment 11: (Results) Please detail the interrater agreement for data extraction.

There was no formal interrater agreement or interrater reliability statistic calculated for this study; instead, we applied consensus rating. We plan to add this to our protocol in future analyses of the ClinicalTrials.gov database. However, we did have a regimented process by which raters were trained in how to label trials. The 6 psychiatrist raters reviewed the list of all of the disorders that fall under each of the DSM-5’s Section II Diagnostic Criteria and Codes (S2 Table in Wortzel et al., 2020)[2]. All 6 psychiatrists then reviewed the same sample of 250 trials to ensure agreement on the labeling criteria. When raters identified any ambiguity when labeling trials, these trials were flagged and reviewed by another psychiatrist. We explain this process in the Materials and Methods section (Tracked version, Lines 103-116). We hope that this description provides clarity to readers of how trials were labeled and how efforts were made to ensure interrater consistency. 

**Comment 12: (Results) Table 1: see my comments about statistical testing.

We chose to keep p-values as a means of interpreting the significance of differences among groups; however, we are now utilizing a more stringent alpha=0.005, as has been recommended in the literature and was suggested by Reviewer 2 [3]. We limit the use of the term ‘significant’ to describing findings with p-values ≤0.005.

**Comment 13: (Results) Please clarify this sentence as the numbers (90.8+23) do not add up to 100 %. I understand that there can be some overlap, but it should be explained: “The top five disorder categories studied (neurodevelopment, substance & addiction, depression, anxiety, and Non-DSM-5 conditions) comprised 90.8% of pediatric mental health trials (Fig 2A). The remaining 14 disorder categories were studied in 23.0% of trials.”

Thank you. We added a sentence after the sentence you highlighted to hopefully provide clarity to why the category percentages sum to greater than 100% (Tracked version, Lines 348-350).

**Comment 14: (Results) Figure 2 is really nice. Congratulations for this figure. Please consider moving “non-DSM” Disorders to the second line (and to order the graph by frequencies).

Thank you – that’s very kind of you to say. We appreciate your advice to move the ‘Non-DSM-5’ category so that the categories are organized from largest to smallest frequency. We made this change.

**Comment 15: (Results) Figure 3 is also a really nice one. For panels B, C and D, I would suggest however to use the same scale for the y-axis. I understand that the number are very different from one category to the other. So perhaps, you can use a log scale. Second possibility, you can use percentages (the numbers being given in panel A).

We have changed the figure so that the y-axis scales for panels Fig 3B-D are now the same. We opted to plot trial frequency for each category rather than percent, as we thought this improved ease of data visualization and interpretation. We decided to keep the y-axis scale linear rather than logarithmic for this reason as well.

**Comment 16: (Discussion) Please insist on the descriptive nature of the study.

We appreciate this feedback. We have added a sentence to the limitations section of the discussion that we hope drives home that this analysis is descriptive in nature (Tracked version, Lines 586-588).

**Comment 17: (Discussion) As there are many comparisons without any adjustment of confounders, please warn explicitly about the fact that no causal interpretations are possible – discuss the issue of confounding thoroughly.

Building off of the sentence added to the limitations section to address your Comment 16, we added an additional sentence reinforcing that there are many comparisons made without adjustments for potential confounders, and causal relationships cannot be drawn from these data (Tracked version, Lines 586-589). We hope that these two sentences together, in addition to the text that comes directly before these added sentences adequately address this concern. The full section addressing the descriptive nature of the analysis, the possibility for confounding, and the inability to draw causal relationships is now as follows (Tracked version Lines 581-591):

“Our study has several possible limitations. First, ClinicalTrials.gov is not an exhaustive list of all US clinical trials [5]. Phase 1 trials and trials studying non-pharmacologic interventions were not subject to the FDAAA or the Final Rule, so these trials may be underrepresented in the registry [6]. There may be other unknown norms and incentives that also bias the registration of certain trial types. Therefore, trends identified in the registry may at least in part reflect changes in trial registration rather than changes in clinical research. Analyses of the registry are also descriptive in nature and include many comparisons that do not account for potential unknown confounders. As a result, causal relationships cannot be drawn from these data. Nevertheless, analyses of ClinicalTrials.gov have allowed many medical specialties to assess trends in clinical research that might otherwise remain unassessable [2, 5, 7-9], as nearly half of trials run by large sponsors go unpublished [10].”

**Comment 18: (Overall) I acknowledge that there are no appropriate reporting guidelines for such meta-research study. But please have a look at equator network and follow the most appropriate guideline (e.g., STROBE?).

Thank you for this guidance. We have reviewed the STROBE guidelines and have added a statement in the Materials and Methods section stating that we adhered to the STROBE reporting guidelines for cross-sectional studies (Tracked version, Lines 248-251).

**Comment 19: (Overall) Please share the data, code and any other information in an adequate repository (e.g., Dryad, https://datadryad.org/stash).

We have uploaded the data and code onto an open-source repository. It can now be found at the following URL: https://datadryad.org/stash/share/8riF8rerFkmSGssL3DaMv4qb07I9W7CC4m3w387zseE

Review 2’s Comments:

**Comment 1: Data availability: The authors should make the dataset used for the analysis available, it is not enough to just say where and how it could be retrieved. The authors might have used criteria or filters or other operations on a large dataset. Readers need to be able to replicate the main analyses. Is there a reason why the authors have opted not to share the extracted data?

Thank you for this feedback. We have now made our data and code available so that the main analysis can be replicated (https://datadryad.org/stash/share/8riF8rerFkmSGssL3DaMv4qb07I9W7CC4m3w387zseE). We were previously unfamiliar with the means by which we could upload this information onto a free server. There is no reason why we cannot share extracted data. Thank you for your guidance with this.

**Comment 2: How does this dataset differ the authors’ previous publication on all mental health research in the same journal (https://journals.plos.org/plosone/article?id=10.1371/journal.pone.0233996)? Weren’t pediatric trials a subset of that? What is the rationale of presenting a separate analysis on that cohort? At least a brief discussion of overlap and the rationale of the more detailed analysis on pediatric trials should be included.

We have added a paragraph to the introduction that explains the extent to which our prior analysis assessed pediatric trials (Tracked version, Lines 66-84). Namely, we previously identified the percentage of US mental health trials that were conducted in the pediatric population, but we did not explore any trends in this trial population. In addition to analyzing trends in specifically pediatric mental health trials, we also manually parsed treatment types in this analysis in a manner more detailed than was performed in the prior analysis in all US mental health trials. In this new paragraph, we also included a review of the literature of prior studies assessing trends in pediatric mental health. We found that, while there have been systematic reviews and meta-analyses of trends in specific treatments in pediatric mental health, there have not been analyses looking at overall trends in pediatric mental health as we set out to do in this study. We hope that this addition to the introduction helps clarify how this study differs from our prior publication (i.e., we think there is little overlap), as well as better identifies the gap this analysis hopes to fill in the literature about trends in pediatric mental health clinical research. 

**Comment 3: Is there any reason for not pre-registering the study protocol? Submission with manuscript, while useful, does not afford the opportunity to assess what was pre-planned and what not.

We agree. Moving forward, we will plan to pre-register our protocols. This seems to be the best practice for ensuring the communication of what was pre-planned and what was changed in our analysis. Unfortunately, that is something that is not possible for the current paper. We do not view this as a major limitation, however, because the focus of our study was largely to review available data for a descriptive analysis of trends rather than primarily an interrogation of a priori hypotheses (for which pre-registration seems more critical). We have made our best effort to detail the changes that were made to our original protocol in two new paragraphs in the Materials and Methods section (Tracked version, Lines 119-153), and we also created a supplemental table (S1 Table) to summarize these changes. In our resubmission we have also included a version of the protocol in which the changes made to the original protocol have been annotated/tracked. We hope that these additions to our submission help provide transparency about our protocol despite the lack of protocol pre-registration. 

**Comment 4: My main objection is about treatment type categories, which appear rather vague and counter-intuitive, mixing type of intervention with delivery type or setting. The authors could consider reorganizing them. For instance, aren’t psychotherapy and pharmacotherapy also interventional? Also “alternative” is an unfortunate label that has a different meaning for many people, as in “alternative” to “conventional” biomedicine. I suggest you replace “Interventional” with “Physical”, as these treatments are often denominated. The “alternative” category is rather heterogenous, mixing types of interventions with modes of delivery or settings. I am also not sure how subcategories fit together, for example there are legitimate, evidence-based technology psychotherapies, whereas yoga or diet have been less studied. Moreover, it also overlaps with psychotherapy, often delivered via the Internet or in other electronic ways. Finally, where do prevention interventions enter?

We greatly appreciated your feedback regarding how to address treatment types. First, we agree that the treatment category “Interventional” is a confusing term. Reviewer 1 suggested that we use the term ‘Stimulation’ instead of ‘Interventional’, and we made this change. Hopefully by removing the term ‘Interventional’ we have eliminated the issue concerning whether psychotherapy and pharmacotherapy are also interventional. In a similar way, we agree that the term ‘Alternative’ caries a connotation of being less conventional/likely less rigorously studied. We have changed this category name to ‘Non-Psycho/Pharmacotherapy’ to more accurately represent the treatments included in this category and to remove this potential connotation. Hopefully now this category is also now more acutely appreciated as a ‘catch-all’ category for interventions that were not medications, psychotherapy, or stimulation.

We appreciate your point that the new category ‘Non-Psycho/Pharmacotherapy’ includes treatments that are unique modes of delivery rather than strictly unique interventions. For example, telepsychiatry might deliver psychotherapy to a patient that, other than being conducted over Zoom in a patient’s home, is no different than the therapy a patient would receive in a clinic. Likewise, a community intervention, such as an assertive community treatment (ACT) program, might provide psychopharmacology treatment within patients’ homes that is similar to that which would be delivered in an intensive outpatient clinic setting. However, these unique modes of delivery can fundamentally change a treatment’s effectiveness. For example, there is evidence that telemedicine interventions for severely mentally ill patients may improve medication adherence and reduce symptom severity and hospitalizations for these patients compared to standard in-person treatment [11]. Psychotic patient enrolled in forensic assertive community treatment programs have significantly fewer criminal convictions and spend less time in jail and in hospitals compared to patients receiving standard outpatient care [12]. Therefore, we think there is merit in identifying these mode of treatment (i.e., telepsychiatry and community programs) as unique treatment subcategories, as they confer distinct treatment outcomes that differ from their conventional in-clinic counterparts. 

However, you brought to our attention a fundamental flaw in how we grouped trials studying treatments involving technology and community. In the original manuscript, the category ‘Technology’ combined trials testing technological treatments, such as videogames and apps using artificial intelligence, with trials that used technology as a mode of treatment (e.g., telepsychiatry). Similarly, the category ‘Community’ grouped trials that tested community programs (e.g., student-led anti-suicide campaigns and afterschool programs) with programs that involved psychiatric care that occurred in the community (e.g., therapists in schools and ACT programs). It is necessary that these types of treatment be distinguished to communicate these differences in how technology and community interventions are used. 

To do this, we reviewed all ‘Technology’ trials and resorted them into ‘Technology’ (i.e., trials using technology as a specific treatment, such as a videogame or interactive app) and ‘Telecommunication’ (i.e., telepsychiatry, etc.). Similarly, we reviewed all ‘Community’ trials and sorted them into ‘Community Programs’ (i.e., afterschool programs, community center activities, etc.) and ‘Community Outreach’ (i.e., ACT programs, integration of mental health into primary care, etc.). We provided an explanation of these new subcategories in the Materials and Methods section (Tracked version, Lines 211-220), and we parse these subcategories in the Results section when they are highlighted (Tracked version, Lines 411-418).

To answer your question regarding the potential for overlap between technology and psychotherapy (e.g., a trial in which CBT is delivered over Zoom), this trial would now be labeled in our study as using both ‘Psychotherapy’ and ‘Telecommunication’. Trials were labeled with as many treatment types as were deemed relevant to what was being studied. In this instance, we deemed that the trial was testing both a psychotherapy treatment as well as a telecommunication treatment. 

To answer your question regarding prevention designation, we determined that ‘prevention’ referred to a trial’s ‘Primary Objective’ (a separate variable provided by ClinicalTrials.gov (Tracked version, Lines 164-173) rather than its treatment type. For example, a trial whose primary objective was to prevent teens from vaping by testing a school-based prevention campaign would have a treatment designation of ‘Non-Psycho/Pharmacotherapy’ with a treatment subcategorization of ‘Community Programs’. We hope this helps answer this question.

In summary, we think that changing the two primary treatment categories to ‘Stimulation’ and ‘Non-Psycho/Pharmacotherapy’, and the addition of two new subcategories under ‘Non-Psycho/Pharmacotherapy’ to parse technology and community treatments, were important alterations to the analysis that strengthen the paper. 

**Comment 5: What is the justification for the alpha threshold? Either the authors employ a correction for multiple comparisons, which is preferable, or at least use the most stringent threshold proposed (p=0.005, see https://www.nature.com/articles/s41562-017-0189-z and associated discussion).

We greatly appreciated your guidance with this and for bringing to our attention the study by Benjamin and colleagues [3]. We updated our protocol and paper to utilize the more stringent alpha threshold of α=0.005. We limit the use of the term ‘significant’ for trends with p-values ≤0.005. 

**Comment 6: I am also not sure all the comparisons in Table 2 make sense or have any relevance, particularly for categories with few trials. It might be more useful to just include statistical comparisons for a few pre-selected categories where there is a reasonable number of included trials, and just present the others descriptively.

Thank you for this point. We agree that comparisons with too few trials makes statistical comparisons within these categories difficult to interpret. For the nine diagnostic categories with fewer than 30 trials, we removed the chi-squared p-values (denoted with dashes in Table 2). We also removed the p-value for the treatment type ‘Stimulation’ in this table, which had fewer than 30 trials. This issue was relevant to the data presented in Table 1, where we also made these changes. We added an explanation in the legends of both tables explaining why p-values are not calculated for these data. As you suggested, we left the data for these categories in the table for descriptive purposes, as these percentage breakdowns are not shown elsewhere in the paper. 

**Comment 7: Please list the relevant R packages used and also consider sharing the code for the analysis, to ensure reproducibility of findings.

We have added the R packages used in the analysis to the Materials and Methods section (Tracked version, Lines 255-256), and we have shared the code for our analysis on a freely accessible server.

**Comment 8: In the introduction, some discussion needs to be included about the fact that increases in prevalence might reflect changes in assessment criteria and instruments.

We appreciate the need for that clarification. We added to the first paragraph of the introduction to provide the caveat that changes in these disorder prevalences may reflect, at least in part, changes in assessment tools and diagnostic accuracy (Tracked version, Lines 39-41).

**Comment 9: Please replace references 44 and 45 with more updated and comprehensive meta-analyses about the similar effectiveness of psychotherapy and pharmacotherapy. There are recent network meta-analyses, individual patient data meta-analyses and so on.

We have replaced these citations with more recent meta-analyses. Thank you for this recommendation.

References:

1. Pica N, Bourgeois F. Discontinuation and Nonpublication of Randomized Clinical Trials Conducted in Children. Pediatrics. 2016;138(3):e20160223. doi: 10.1542/peds.2016-0223. PubMed PMID: 27492817.

2. Wortzel JR, Turner BE, Weeks BT, Fragassi C, Ramos V, Truong T, et al. Trends in mental health clinical research: Characterizing the ClinicalTrials.gov registry from 2007–2018. PLOS ONE. 2020;15(6):e0233996. doi: 10.1371/journal.pone.0233996.

3. Benjamin DJ, Berger JO, Johannesson M, Nosek BA, Wagenmakers EJ, Berk R, et al. Redefine statistical significance. Nature Human Behaviour. 2018;2(1):6-10. doi: 10.1038/s41562-017-0189-z.

4. Tse T, Fain KM, Zarin DA. How to avoid common problems when using ClinicalTrials.gov in research: 10 issues to consider. BMJ. 2018;361. doi: 10.1136/bmj.k1452.

5. Liu X, Zhang Y, Tang L, et al. Characteristics of radiotherapy trials compared with other oncological clinical trials in the past 10 years. JAMA Oncology. 2018. doi: 10.1001/jamaoncol.2018.0887.

6. Tse T, Fain KM, Zarin DA. How to avoid common problems when using ClinicalTrials.gov in research: 10 issues to consider. Bmj. 2018;361:k1452. Epub 2018/05/29. doi: 10.1136/bmj.k1452. PubMed PMID: 29802130; PubMed Central PMCID: PMCPMC5968400 declaration of interests and declare the following interests: none.

7. Pasquali SK, Lam WK, Chiswell K, Kemper AR, Li JS. Status of the pediatric clinical trials enterprise: an analysis of the US ClinicalTrials.gov registry. Pediatrics. 2012;130(5):e1269-77. Epub 2012/10/03. doi: 10.1542/peds.2011-3565. PubMed PMID: 23027172; PubMed Central PMCID: PMCPMC4074644.

8. Arnow KD, King AC, Wagner TH. Characteristics of mental health trials registered in ClinicalTrials.gov. Psychiatry Res. 2019;281:112552. Epub 2019/10/19. doi: 10.1016/j.psychres.2019.112552. PubMed PMID: 31627072.

9. Califf RM, Zarin DA, Kramer JM, Sherman RE, Aberle LH, Tasneem A. Characteristics of clinical trials registered in clinicaltrials.gov, 2007-2010. JAMA. 2012;307(17):1838-47. doi: 10.1001/jama.2012.3424.

10. Iacobucci G. Nearly half of all trials run by major sponsors in past decade are unpublished. BMJ. 2016;355:i5955. doi: 10.1136/bmj.i5955.

11. Lawes-Wickwar S, McBain H, Mulligan K. Application and Effectiveness of Telehealth to Support Severe Mental Illness Management: Systematic Review. JMIR Ment Health. 2018;5(4):e62. Epub 21.11.2018. doi: 10.2196/mental.8816. PubMed PMID: 30463836.

12. Lamberti JS, Weisman RL, Cerulli C, Williams GC, Jacobowitz DB, Mueser KT, et al. A Randomized Controlled Trial of the Rochester Forensic Assertive Community Treatment Model. Psychiatric Services. 2017;68(10):1016-24. doi: 10.1176/appi.ps.201600329.

---

## [Decision Letter · Decision Letter 1]

8 Mar 2021

Trends in US pediatric mental health clinical trials: An analysis of ClinicalTrials.gov from 2007 – 2018

PONE-D-20-34754R1

Dear Dr. Wortzel,

We’re pleased to inform you that your manuscript has been judged scientifically suitable for publication and will be formally accepted for publication once it meets all outstanding technical requirements.

Kind regards,

Claudio Gentili

Academic Editor

PLOS ONE

Additional Editor Comments (optional):

Reviewers' comments:

Reviewer's Responses to Questions

**Comments to the Author**

1. If the authors have adequately addressed your comments raised in a previous round of review and you feel that this manuscript is now acceptable for publication, you may indicate that here to bypass the “Comments to the Author” section, enter your conflict of interest statement in the “Confidential to Editor” section, and submit your "Accept" recommendation.

Reviewer #1: All comments have been addressed

Reviewer #2: All comments have been addressed

2. Is the manuscript technically sound, and do the data support the conclusions?

Reviewer #1: Yes

Reviewer #2: Yes

3. Has the statistical analysis been performed appropriately and rigorously? 

Reviewer #1: Yes

Reviewer #2: Yes

4. Have the authors made all data underlying the findings in their manuscript fully available?

Reviewer #1: Yes

Reviewer #2: Yes

5. Is the manuscript presented in an intelligible fashion and written in standard English?

Reviewer #1: Yes

Reviewer #2: Yes

6. Review Comments to the Author

Reviewer #1: Only comment : as far as it is possible move the paragraph CHANGE TO THE INITIAL PROTOCOL at the end of the method section or at the start of the results section

Reviewer #2: There are some spelling errors in the sections added, e.g. "systemic" instead of "systematic" in discussing the systematic reviews in the Introduction.

7. PLOS authors have the option to publish the peer review history of their article (what does this mean?). If published, this will include your full peer review and any attached files.

Reviewer #1: **Yes: **Florian NAUDET

Reviewer #2: **Yes: **Ioana A. Cristea

---

## [Editor Report · Acceptance letter]

22 Mar 2021

PONE-D-20-34754R1 

Trends in US pediatric mental health clinical trials: An analysis of ClinicalTrials.gov from 2007 – 2018 

Dear Dr. Wortzel:

I'm pleased to inform you that your manuscript has been deemed suitable for publication in PLOS ONE. Congratulations! Your manuscript is now with our production department. 

Kind regards, 

on behalf of

Professor Claudio Gentili 

Academic Editor

PLOS ONE